# DNA Sequence Analysis of an Inversion Hot Spot in Lobeliaceae Plastomes

**DOI:** 10.3390/plants11212863

**Published:** 2022-10-27

**Authors:** Eric B. Knox

**Affiliations:** Department of Biology, Indiana University, Bloomington, IN 47405, USA; eknox@indiana.edu; Tel.: +1-812-855-9601

**Keywords:** *accD*, disruption-rescue inversion model, DNA trafficking, duplicative transposition, *rpl23*, stem-loop inversion model, stem-loop ligation model

## Abstract

The evolution of plastid genomes (plastomes) in land plants is typically conservative, with extensive structural rearrangements present in only a few groups. Early Southern blot analysis identified two *Lobelia* species that minimally required deletion of the plastid gene *accD* and five inversions to account for their plastome arrangement relative to the ancestral organization. Sixty alternative 5-step inversion scenarios could account for the observed arrangement, but only one scenario was consistent with the criterion of ‘common cause’ attributable to a putative rearrangement hot spot at the *accD* deletion-site. Plastome sequencing demonstrated that this previously hypothesized inversion order is historically accurate. Detailed reconstructions of the ancestral plastome organization before and after each inversion are presented herein. Stem-loop and disruption-rescue models were evaluated for each inversion. One inversion has an obvious stem-loop basis, but the other four inversions were primarily caused by serial insertion of foreign (extra-plastid) DNA bearing large open-reading frames that disrupted plastome organization at the *accD* deletion-site, and complete plastomes were rescued by seemingly arbitrary ligation or fortuitous recombination at the other inversion endpoint. Transposed copies of DNA segments from elsewhere in the plastome are frequently inserted at inversion junctions, and four junctions are consistent with the stem-loop ligation model.

## 1. Introduction

Prior to the advent of DNA sequencing, plant molecular systematics relied on restriction-site analysis of plastid genomes (plastomes) to infer phylogenetic relationships [1]. Widespread adoption of this method quickly revealed the conservative nature of plastome evolution in most land plants, with a few groups noted for their extensive structural rearrangements [2]. The Lobeliaceae are one such group that showed a surprising diversity of plastome arrangements that were attributable to deletions and large inversions, the distribution of which was congruent with the phylogenetic relationships inferred from the restriction-site analysis [3]. Southern blot probing of restriction enzyme digests provided effective assessment of large-scale rearrangement patterns but had obvious limitations at finer resolution. For example, the probing results appeared to show independent partial deletions of *clpP* in *Lobelia holstii* and *Monopsis lutea* [3], but the lack of probe hybridization actually resulted from the independent loss of both *clpP* introns, coupled with rapid sequence divergence of the coding region [4].

The methodological shift to plastome gene sequencing provided a more effective community-based approach to estimating plant phylogeny [5], but gene sequences alone provide no insight into overall plastome organization and evolution. Early Sanger sequencing of complete plastomes from diverse plant species helped identify the conserved open-reading frames (ORFs; labeled as hypothetical chloroplast open-reading frames; = ycf genes) whose functions in all but one case (*ycf2*) has subsequently been identified. However, the intergenic regions of these scattered exemplars were generally too divergent for comparative analysis. Multiplexed ‘genome skimming’ with next-generation sequencing [6,7] provides a cost-effective approach to recovering complete plastomes in closely related species, and will usher in a new era of exceptionally well supported phylogenetic results plus the comparative evidence needed to characterize and understand the evolutionary properties of this workhorse genome.

Of the *Lobelia* species originally studied in 1993, *L. fervens* and *L. erinus* had the most extensively rearranged plastomes. The derivation of this rearrangement pattern from a tobacco-like ancestral pattern [8] minimally required deletion of *accD* plus five large inversions. All Lobeliaceae lack *accD*, so this gene deletion was safely inferred to have been an early event (and is now known to have occurred in the common ancestral lineage with the closely related Campanulaceae and Cyphiaceae; [4]; Figure 1). Evaluation of all 5-step inversion models yielded 5!/2 = 60 alternative scenarios because one of the inversions must have overlapped a previous inversion. The admittedly limited phylogenetic evidence available in 1993 suggested that the first inversion was shared with *Monopsis lutea* and the second inversion was shared with *L. cardinalis* and the giant lobelias (which were the focus of the study), but the remaining three inversions could have occurred in various possible orders. If the first two inversions were synapomorphic with other Lobeliaceae, then the *accD* deletion-site was a common endpoint for both these inversions, and of the remaining alternative scenarios, only one consistently used the *accD* deletion-site as an endpoint for all five inversions. This additional criterion of ‘common cause’ was the basis for the hypothesized order of rearrangement events (see Figure 4 in [3]), which suggested the inversions were not random events, but resulted from some underlying molecular factor responsible for a recombinational hot spot [9]. Phylogenetic analysis of complete plastome sequences demonstrated that the previously hypothesized order of inversions is the historically accurate explanation for the plastome arrangements observed in *L. fervens* and *L. erinus* [4] because Inversions 1, 2, and 5 occur in different phylogenetic intervals (Figure 1). The order of Inversions 3 and 4 are known because the endpoint of Inversion 4 was offset from the hot spot, and the position and orientation of an approximately 340-base-pairs (bp) segment of intergenic plastid DNA is plausibly explained only if Inversion 3 preceded Inversion 4. This paper reconstructs the underlying molecular factors responsible for these five inversions.

Typical land plant plastomes have a quadripartite structural organization, with two copies of the inverted repeat (IR) region that separate the large single-copy (LSC) and small single-copy (SSC) regions (see Figure 1 in [4]), but they are functionally tripartite because the IR evolves as a single unit, which recombines with sufficient frequency to maintain equimolar populations of molecules with the single-copy regions inverted relative to one another [10,11]. Hence, IR-containing plastomes function like a double-headed, stem-loop structure (with the IR forming the stem and the single-copy regions forming the loops at either end), and inversions at this scale occur very, very frequently as the two IR copies recombine during their concerted evolution. At the smallest possible scale, hairpin inversions are common [12] (see also Figure S3B in [4]), which occur because the flanking sequences are palindromic (comprising reverse and complementary nucleotides) and can form miniature stem-loop structures. Each inversion has two endpoints, and a stem-loop model accounts for the 34-kilobase (kb) inversion in the LSC of *Lobelia hartlaubii* relative to *L. baumannii*, which is flanked by an inverted, imperfect, 1-kb repeat that undergoes concerted evolution (see Figure 3 in [4]), and the independent 5-kb inversions in the IR of *L. heterophylla* and *L. linearis*, which is flanked by a 15-bp stem (see Figure S3C in [4]). This stem-loop model for inversion is thus characterized by flanking sequences in inverted orientation, with the potential for recombination increasing with the size of the stem and decreasing with the length of the loop. This symmetrical stem-loop model will be compared with an asymmetrical disruption-rescue model, in which plastome organization is disrupted by an event at one inversion endpoint, but an intact molecule is rescued by an unrelated ligation event at the other inversion endpoint. In addition to the extensive inversions, Lobeliaceae plastomes were repeatedly invaded by inserted segments of foreign (extra-plastid) DNA that carry large ORFs [4], and this paper evaluates whether foreign DNA insertions caused one or more of the inversions in the lineage leading to *L. fervens* and *L. erinus*.

## 2. Results

### 2.1. Deep Ancestry with the Campanulaceae and Cyphiaceae

The five inversions shared by *Lobelia fervens* and *L. erinus* illustrate the complex alterations that have occurred at inversion and insertion junctions. Not all historical details can be unambiguously reconstructed because of dramatic subsequent alterations in descendent lineages. Broader comparison with the Campanulaceae and Cyphiaceae (Figure 1) indicate that in addition to the loss of *accD* from the common ancestral plastome, there was also a duplicative transposition of *rpl23* (from the IR) to the intergenic spacer between *trnC(GCA)* and *rpoB* (located in the LSC; [4]). This copy of *rpl23* was almost complete and potentially functional because of an in-frame stop codon just downstream from the insertion site. The transposed segment begins upstream of *rpl23* at the base of a small stem-loop structure, and the insertion site in the *trnC(GCA)*-*rpoB* region had seven of eight nucleotides that coincidentally matched the sequence flanking that stem-loop structure. Insertion of the transposed copy of *rpl23* is inferred to have occurred via a single-stranded, stem-loop ligation model (Figure 2) at the upstream junction and an arbitrary ligation point at the downstream junction. The duplicated start of *rpl23*, along with the flanking 64-bp of *trnC(GCA)-rpoB* intergenic DNA, was secondarily transposed to a site downstream of *rbcL* (also in the LSC; Figure 3). This secondarily transposed copy is flanked on both sides by foreign DNA (so the mechanism of integration cannot be determined at this time), but the first two codons of the *rpl23* copy start the inferred ancestral ORF200, which was inserted into the common ancestral plastome (Figure 1). The very short phylogenetic interval of the common ancestral lineage of the Campanulaceae and Cyphiaceae (Figure 1) provides evidence that *accD* was already reduced to remnants in the ancestral lineage of the Lobeliaceae (Figure 3), but it is not possible to determine what, if any, sequence was present between ORF200 and the *accD* remnants.

### 2.2. Five Successive Inversions

#### 2.2.1. Inversion 1

The inversions in the *Lobelia baumannii-Colensoa* clade occurred independently from the inversions in the rest of the Lobeliaceae, the first of which occurred in the common ancestral lineage of *Monopsis* and the remaining Lobeliaceae (Inversion 1 in Figure 1). Associated with this inversion are (1) a 53-bp deletion from the *psbA*-*trnK(UUU)* intergenic region, (2) a 118-bp duplication of the 3′ end of ORF200 (now forming an inverted, imperfect, dispersed repeat), (3) insertion of a large segment of foreign DNA inferred to have carried ancestral ORF148, (4) a 64-bp copy of the *trnC(GCA)-rpoB* intergenic spacer, and (5) a 45-bp tandem duplication of plastid DNA at the inversion endpoint upstream of *psbA* (Figure 3). It is possible that the 118-bp 3′ copy of ORF200 was transposed (in inverted orientation) into the *psbA*-*trnK(UUU)* intergenic region, where it then provided the molecular basis of a stem-loop inversion, but it seems more likely that this segment was duplicated during the plastome disruption caused by the insertion of ORF148. This insertion, along with the seemingly arbitrary ligation points in the *psbA*-*trnK(UUU)* intergenic region, which resulted in the 53-bp deletion and the 45-bp tandem duplication, is more consistent with the disruption-rescue model. It is possible that the ligation points represent sites of illegitimate recombination due to sequence similarity, but this hypothesis cannot be evaluated without knowledge of the foreign DNA sequences prior to integration into the plastome. The duplicative transposition of the 64-bp segment from the *trnC(GCA)*-*rpoB* region and the deletion that removed almost all remaining traces of *accD* may have occurred at the same time as the inversion, or they may have occurred at a later time during the same phylogenetic interval. The plastome of *L. thermalis* (Figure 1) has only this one inversion in the LSC (plus two unique inversions in the IR). The inverted, imperfect, 118-bp repeat evidently served as a stem-loop reversion site in *Grammatotheca bergiana* (Figure 1), with no associated deletions, duplications, or insertions (Figure 3). The shared features with *L. thermalis* at the inversion junctions demonstrate that this was a reversion, not retention of the unrearranged ancestral genome organization. Neither the inferred ancestral ORF200 nor ORF148 is preserved intact in any of the completely sequenced plastomes, but many species preserve large segments of foreign DNA, from which the ancestral coding frames can be reconstructed.

#### 2.2.2. Inversion 2

The inversion that occurred in the common ancestral lineage of *Lobelia galpinii* and the remaining Lobeliaceae (Inversion 2 in Figure 1) has an associated insertion of foreign DNA carrying ORF227 that replaced the segment that was previously located at the inversion hot spot and a 152-bp deletion from the *trnV(UAC)*-*ndhC* intergenic region (Figure 4). A small, pre-existing, stem-loop structure upstream of *ndhC* now flanks one inversion endpoint. This junction with a transposed copy of the 3′ end of *rpl23* is spanned by an enlarged stem, which is consistent with the stem-loop ligation model (Figure 2). The junction of the 3′ end of *rpl23* with the foreign DNA carrying ORF227 cannot be evaluated without knowing the foreign source sequence. Weak amino acid similarity between the 3’ end of ancestral ORF227 and the 118-bp remnant of ancestral ORF200 raises the possibility that the DNA sequences were similar enough to permit recombination, but all remnants of *accD* and the flanking sequence were deleted and the inserted foreign DNA now flanks a stem-loop structure upstream of *psaI* (Figure 4). The *trnV(UAC)-trnK(UUU)* junction lacks any obvious sequence element that was involved in the inversion. The small copied segment of the *trnK(UUU)* intron may be the remnant of a larger tandem duplication, which implies that the original junction with the *trnV(UAC)* downstream intergenic DNA was located somewhere downstream of this *trnK(UUU)* intron copy. A subsequent large deletion would account for the position and orientation of the *trnK(UUU)* intron copy, the missing 152 bp from the *trnV(UAC)*-*ndhC* intergenic region, the missing end of the hypothesized tandem duplication, and whatever DNA was previously present downstream of *trnK(UUU)*. However, the abutment of the *trnV(UAC)* downstream intergenic DNA and the duplicated copy of the *trnK(UUU)* intron is not consistent with a slipped strand deletion event, and there are no matching sequences in the relevant ancestral *trnV(UAC)*-*ndhC* and *trnK(UUU)*-*psaI* regions that would have facilitated an inversion, so no ancillary evidence supports this hypothesis. If the original *trnV(UAC)-trnK(UUU)* junction is preserved, then this inversion is consistent with the disruption-rescue model, and the location of the transposed *trnK(UUU)* intron copy so close to the source is just coincidence.

#### 2.2.3. Inversion 3

The order of the two inversions shared by the *Lobelia fervens*-*L. laxa* clade (Inversions 3 and 4 in Figure 1) can be inferred because the inversion endpoints are offset. Plastome sequences in species outside this clade clearly record the complex history of the *rpoB*-*trnC(GCA)* intergenic region (Figure 5). An almost complete (and potentially functional) copy of *rpl23* was transposed in the plastome of the common ancestral lineage of the Campanulaceae, Cyphiaceae, and Lobeliaceae (Figure 1). Most of this *rpl23* copy and the flanking segment of intergenic plastid DNA was subsequently replaced by foreign DNA that incorporated a duplicated copy of the start of *ycf1* into the ancestral ORF308 (Figure 5). One junction of Inversion 3 was formed by recombination of the remnant copy of the 3′ end of *rpl23* located upstream of *rpoB* and the copy of the 3′ end of *rpl23* that was incorporated upstream of *ndhC* during Inversion 2 (Figure 4). The other inversion junction has a transposed copy of *rps12* intron 1B located upstream of foreign DNA that carries ORF128, and these newly inserted segments are flanked by a highly modified ORF236 (potentially derived from the ancestral ORF308) and a small remnant of ancestral ORF227. The *rps12* intron 1B copy has five recently derived point mutations that are shared with the source region in *L. laxa*, which indicates that this was a very recent duplicative transposition in *L. laxa* or that the copy and the source region are undergoing concerted evolution. There is weak amino acid similarity among small segments at the 3′ ends of ORF236, ancestral ORF308, and ancestral ORF227, but if ORF236 is a recombination product of ancestral ORF308 and ancestral ORF227, then a very complex set of events must be hypothesized to account for the ORF227 remnant located farther downstream. This inversion is more consistent with the disruption-rescue model, where the rescue was accomplished by recombination of the two copies of the 3′ end of *rpl23* into a single copy. In the stem-loop model of inversion, both stems are preserved. *Lobelia laxa* is the only known species that is sister to the *L. fervens*-*L. erinus* clade, so there is no immediate phylogenetic opportunity to investigate the timing of events in the *L. laxa* lineage.

#### 2.2.4. Inversion 4

In contrast to the complexity of Inversion 3, Inversion 4 involved an almost trivial match of seven bp in the *trnG(UCC)*-*trnR(UCU)* and *trnC(GCA)*-*psaI* intergenic regions, with no associated insertions, deletions, or duplications (Figure 6). This is clearly a stem-loop inversion, and while the matching sequences were obviously necessary for this inversion, such matching is not sufficient to explain why this inversion occurred. Although Inversions 3 and 4 occurred in the same phylogenetic interval, the orientation and location of blocks C and D from the *rpoB*-*trnC(GCA)* intergenic region (Figure 5) after the fourth inversion (Figure 6) provides compelling evidence that Inversion 3 preceded Inversion 4.

#### 2.2.5. Inversion 5

The inversion shared by the *Lobelia fervens*-*L. erinus* clade (Inversion 5 in Figure 1) displays many features consistent with the disruption-rescue model. The *rps4-trnT(UGU)* intergenic region in *L. laxa* has insertions of foreign DNA and transposed copies of plastid DNA (Figure 7) that can be dated by a secondary transposition of the 3′ *rpl23* remnant and the flanking foreign DNA from the *rpoB*-*trnC(GCA)* intergenic region (Figure 5) that was deleted from the plastome of the common ancestral lineage of the *L. fervens*-*L. laxa* clade. One junction of Inversion 5 was formed by recombination of the large stem-loop structure upstream from *psaI* and a matching 13-bp segment that was already present in the *rps4-trnT(UGU)* intergenic region. The stem-loop structure may have promoted recombination at this site, but this was not a stem-loop inversion. The other inversion junction has a newly formed stem-loop at the end of the intergenic plastid DNA upstream of *rps4* and extensive deletions and insertions that have continued after divergence of the *L. fervens* and *L. erinus* lineages. As a result of this DNA trafficking, only a 300-bp core in ORF180 and ORF188 remains from ORF236, and the remaining foreign DNA and some of the flanking plastid intergenic DNA from the *rps4-trnT(UGU)* and *trnR(UCU)-psaI* regions has been replaced by new insertions. Because ORF125 is intact in *L. fervens* and some plastid DNA from the *rpoB*-*trnC(GCA)* intergenic region (Figure 5) is still present upstream of *trnR(UCU)* in this species (following Inversion 4), the lineage leading to *L. erinus* is inferred to have sustained more extensive and recent alterations. However, DNA trafficking also appears to have continued in the lineage leading to *L. fervens*, as evidenced by the transposed copy of *rps12* intron 1B and the flanking intergenic plastid DNA. The source of this duplicative transposition overlaps the source of the *rps12* intron 1B copy in *L. laxa*, but the location of the copy is different. The *rps12* intron 1B copy in *L. fervens* has four recently derived point mutations that are shared with the source region in *L. fervens*, which indicates that this was a very recent duplicative transposition in *L. fervens* or that the copy and the source region are undergoing concerted evolution. *Lobelia fervens* and *L. erinus* are part of a large, rapidly diversifying clade that provides ample phylogenetic opportunity to study the most recent alterations by sequencing the plastomes of related species.

## 3. Discussion

With an appropriate phylogenetic context, ancestral DNA sequences can be reconstructed at the beginning and end of each phylogenetic interval. When evolutionary changes in the descendent lineages are relatively modest, ancestral sequences are confidently reconstructed and DNA ambiguity codes adequately represent the few positions at which a nucleotide differs between sister clades and these differ from the nucleotide present at the start of a phylogenetic interval. Although alignment gaps are commonly referred to as ‘indels’ (shorthand for insertions/deletions), bona fide insertions are rare in angiosperm plastomes [4] and a better name would be ‘dupdels’ because most so-called insertions are actually short tandem-duplications. The differences between the reconstructed ancestral sequences at the beginning and end of each phylogenetic interval are due to evolutionary changes during that interval, the temporal order of which cannot be determined in most cases.

The inserted foreign DNA in the Lobeliaceae (and Campanulaceae and Cyphiaceae) is commonly associated with transposed copies of DNA from elsewhere in the plastome (Figure 3, Figure 4, Figure 5 and Figure 7), and many of the foreign ORFs are chimeric constructs that have ‘hijacked’ a copy of the upstream sequence and start of a plastid gene (see Table S3 in [4]). In the absence of other evidence, it is parsimonious to attribute such a heterogeneous DNA segment to a single, complex insertion event, but the Lobeliaceae also show clear evidence of DNA trafficking at many sites in the plastome, and such heterogeneous segments in some cases are the result of sequential insertions and replacements (e.g., Figure 7).

The fact that extant species preserve three of the four inferred intermediate plastome arrangements from the hypothesized 5-step inversion order [3,4] bolsters confidence that the actual history of Lobeliaceae plastome evolution is reasonably approximated by rational reconstruction of ancestral DNA sequences and rearrangement events. The offset endpoints of Inversions 3 and 4 left an approximately 340-bp segment of the *rpoB*-*trnC(GCA)* intergenic region in a position and orientation that is plausibly explained only if Inversion 3 preceded Inversion 4. This small segment is preserved in *Lobelia laxa* (Figure 6) but is partially deleted in *L. fervens* and completely deleted in *L. erinus* (Figure 7). *Lobelia fervens* and *L. erinus* triangulate the base of a large clade of predominantly African blue-flowered species, but *L. laxa* is the only known relative to this clade. As a result, the extant *L. laxa* provides the only evidence of what may have been present upstream of *psaI* after Inversion 3 (Figure 5) and in the *rps4-trnT(UGU)* region prior to Inversion 5 (Figure 7). Although there is no obvious avenue for obtaining additional evidence about this node by expanded phylogenetic sampling, there are relatives of *L. thermalis*, *L. sonderiana*, and *L. galpinii* (Figure 1) that do not yet have complete plastomes, and additional sampling may strengthen and/or refine our understanding of the molecular factors responsible for Inversions 1 and 2.

Not all foreign DNA insertions caused inversions, and not all inversions were caused by foreign DNA insertions. The ancestral ORF200 is inferred to have been inserted at or near the *accD* deletion-site (Figure 3) without causing an inversion (Figure 1). Similarly, insertion of ancestral ORF308 in the *trnC(GCA)*-*rpoB* intergenic region (Figure 5) and other ORFs at other sites in the plastome (see Table S3 in [4]) did not cause inversions. However, four of the five inversions presented here have associated foreign DNA insertions, and it seems likely in these cases that the insertions disrupted the plastome organization and ultimately caused these inversions. The plausibility of this disruption-rescue model is bolstered by the frequent asymmetrical pattern at the inversion endpoints, wherein one junction has dramatic changes and the other junction has seemingly arbitrary ligation points or fortuitous recombination, sometimes with associated deletion or replacement of DNA segments known to have been present in the ancestral lineage. It is possible that additional disruptive insertions previously occurred, but the only plastomes to survive such events were those that were ‘rescued’ by ligation at the second junction, thereby preserving a complete plastome.

The reversion in *Grammatotheca bergiana* (Figure 3) has a more obvious and conventional molecular basis, namely the inverted 118-bp duplicated sequence from the 3′ end of ancestral ORF200. These copies were separated by more than 56 kb of intervening sequence, but such dispersed, inverted, repeats potentially act like ‘molecular velcro’ in promoting recombination. The frequency with which such stem-loop inversions occur likely depends on the length of the duplicated sequence (the stem) and the distance separating the copies (the loop). At the largest possible scale, recombination between the IR copies is sufficiently frequent to maintain equimolar populations of the two structural isomers [10,11]. At the smallest possible scale, hairpin inversions [12] are common because of the close proximity of the palindromic stems. The inverted, imperfect, 1-kb repeat responsible for the 34-kb inversion in *Lobelia hartlaubii* relative to *L. baumannii* undergoes concerted evolution, and it is likely that the intervening 34-kb has flip-flopped repeatedly [4]. *Lobelia heterophylla* and *L. linearis* have independent 5-kb inversions in the IR caused by inverted 15-bp segments that coincidentally match [4], but the frequency of independent inversions with such small stems and large loops is predicted to be low. Inversion 4 (Figure 6) occurred when inverted seven-bp segments, located over 40 kb apart, recombined. Although it is obvious how this inversion occurred, it is not clear why this inversion occurred. The matching sequences provided a necessary molecular basis for the inversion, but such matching is not sufficient to explain why this inversion occurred because such seven-bp matches are common in plastomes, and large inversion would occur frequently if such matches were the entire mechanistic explanation.

Transposed copies of DNA from elsewhere in the plastome provide important historical information. Secondary transpositions that overlap a junction from a primary transposition (or subsequent insertion; cf. Figure 3, Figure 4, Figure 5 and Figure 7) preserve evidence of that original junction, even though the original junction may have been subsequently deleted in one of the descendant clades (e.g., Figure 5). These secondary transpositions allow the temporal order of events to be determined. Primary transpositions can also help date events by including the transposed copy in a phylogenetic analysis of the source region, but such results must be interpreted cautiously because a high level of sequence similarity could be due to concerted evolution instead of very recent duplicative transposition. The frequency of such concerted evolution is predicted to increase with the length of the transposed copy.

Primary and secondarily transposed partial copies of *rpl23* play recurrent roles in the plastome rearrangements, being one of the early shared events in the ancestral lineage with the Campanulaceae and Cyphiaceae (Figure 1 and Figure 5), providing the chimeric start for ancestral ORF200 (Figure 3), and serving as a recombinational ‘rescue unit’ for Inversion 3 (Figure 5). Unlike the examples of stem-loop inversions (discussed above), the two dispersed copies of the 3′ end of *rpl23* recombined into a single copy and cannot be considered the primary cause of Inversion 3.

Small stem-loop structures also appear to have a recurrent role in plastome rearrangements, both as breakage points and ‘molecular jigs’ for ligation (Figure 2). The presence of a rearrangement junction in an expanded palindromic sequence provides compelling evidence that some version of stem-loop ligation has occurred. For junctions involving transposed copies of plastid DNA joined to other segments of plastid DNA (e.g., Figure 4 and Figure 5), the ancestral sequences can be reconstructed with high confidence.

Without knowing the source sequence of the inserted segments of foreign DNA it is not possible to analyze fully their junctions with plastid DNA. Comparative plastome mapping delimits the endpoints of the plastid DNA, beyond which the DNA is unambiguously of extra-plastid origin. However, it is possible that a foreign DNA segment started with sequence that coincidentally matched the plastid DNA, with illegitimate recombination and/or stem-loop ligation partially accounting for how the foreign DNA integrated into the plastome. When stem-loop structures form or are enlarged at plastid/foreign junctions (e.g., Figure 7), circumstantial evidence suggests that the underlying mechanisms are similar to plastid/plastid junctions, but direct evidence is lacking.

The source of the inserted foreign DNA is not yet known but is most likely the nucleus [4]. The previous plastome sequencing focused on the early evolutionary history of the Lobeliaceae. Broad surveys of closely related species are necessary to identify the most recent cases of foreign DNA insertions because recent cases will retain more evidence of how and why these insertions are occurring. The recent DNA trafficking that is apparent in the comparison of *Lobelia fervens* and *L. erinus* suggests that more extreme alterations within this clade remain to be discovered—perhaps species that have additional inversions. Four of the five inversions documented herein are consistent with the disruption-rescue model, with one endpoint located at the inversion hot spot located at the *accD* deletion-site. However, the “*accD* deletion-site” describes what is not present, not what is present. This inversion hot spot is characterized by serial replacement of foreign DNA at a site upstream of *psaI* that is adjacent to a large stem-loop structure. The quintessential feature of this hot spot seems to be the foreign DNA trafficking, but to what extent is the adjacent stem-loop structure involved? Inversion 5 separated these two features (Figure 7), and the location of any additional inversions in the *L. fervens*-*L. erinus* clade may answer this question.

## 4. Materials and Methods

Inferred ancestral sequences were reconstructed from multiple sequence alignments of previously sequenced plastomes [4]. Each inversion has two junctions. For each phylogenetic interval during which an inversion occurred, the descendent lineages were used to reconstruct the sequence of the inversion junctions at the end of that interval. Related lineages were compared to infer the ancestral sequence present at the start of each interval. Using basic phylogenetic reconstruction techniques for inferred ancestral character states, these ancestral sequences were included in the multiple sequence alignments using the editing features of Sequencher (Gene Codes Corporation, Ann Arbor, MI, USA), and parsimony analysis [13] was used to confirm their accuracy. In relatively few ancestral DNA sequence positions, DNA ambiguity codes were used to represent possible ancestral character states that could not be unambiguously inferred from the available sequences. Comparative mapping determined the endpoints of the plastid DNA segments. All extra-plastid DNA segments are drawn to represent their minimum possible extent because extra-plastid sequence similar to plastid DNA cannot be distinguished from bona fide plastid DNA. Transposed copies of plastid DNA were compared with the source regions to estimate the relative recency of each duplicative transposition and for evidence of concerted evolution. Stem-loop structures in sequenced plastomes were readily identified in low-stringency Blast [14] results and by inverted loops in multiple sequence alignments (see Figure S3B in [4]). Similar stem-pairing potential was determined in the reconstructed ancestral sequences.

## Figures and Tables

**Figure 1 plants-11-02863-f001:**
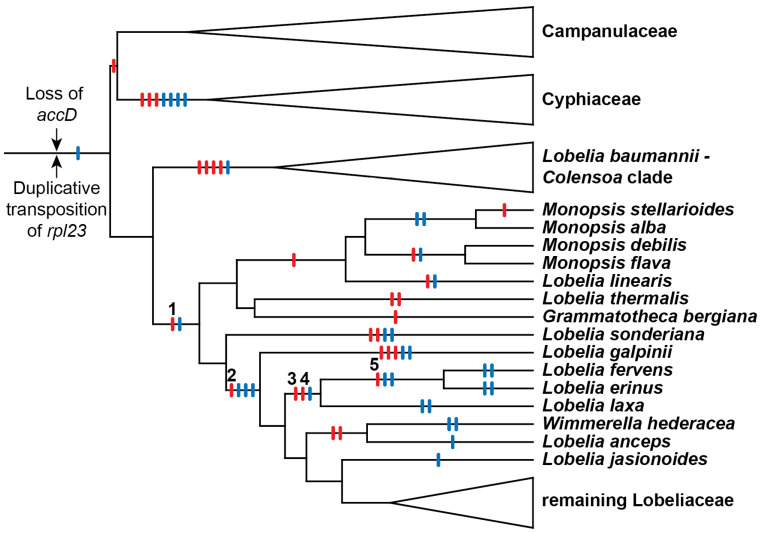
Phylogenetic context for the five plastome inversions (numbered red hash marks) in the lineage leading to *Lobelia fervens* and *L. erinus* (redrawn from [4]). The temporal order of the *accD* gene loss and duplicative transposition of *rpl23* in the common ancestral lineage of the Campanulaceae, Cyphiaceae, and Lobeliaceae cannot be determined, but insertion of the ancestral ORF200 (blue hash marks indicate the insertion of foreign ORFs) is inferred to have occurred subsequently. *Lobelia thermalis* only has Inversion 1 in the LSC (the two subsequent inversions are in the IR), *L. jasionoides* (and many other species) only have Inversions 1 and 2, and *L. laxa* has four of the five inversions.

**Figure 2 plants-11-02863-f002:**
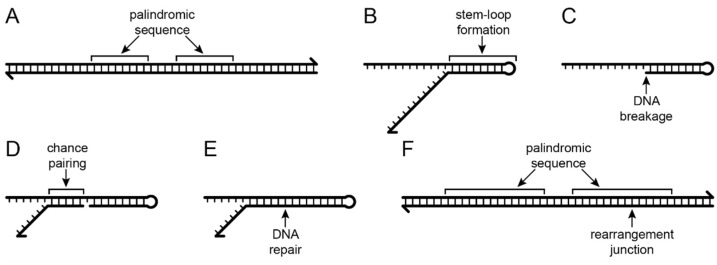
Stem-loop ligation model. (**A**) Plastomes typically have many small palindromic sequences that are the reverse and complement of one another. (**B**) When DNA strands are separated, the complementary nucleotides can form stem-loop structures. (**C**) The base of the stem may be prone to DNA breakage. (**D**) Chance pairing may provide a ‘molecular jig’ that temporarily stabilizes a new DNA junction. (**E**) DNA repair ligates the junction. (**F**) The rearrangement junction is located in an expanded palindromic sequence.

**Figure 3 plants-11-02863-f003:**
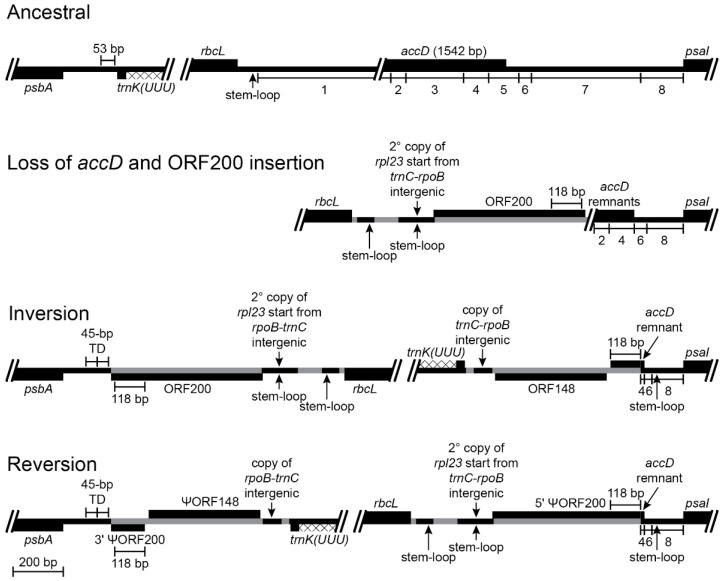
Reconstruction of the *psbA*-*trnK(UUU)* and *rbcL*-*psaI* regions before and after an inversion and the associated insertions of foreign DNA and transposed copies of plastid DNA (plastid DNA is depicted as a black line and inserted segments of foreign DNA are marked in gray; genes depicted above the line are transcribed left-to-right; genes depicted below the line are transcribed right-to-left; protein coding regions, tRNA genes, and foreign ORFs are depicted as black boxes; introns are marked with a cross-hatched pattern; bp = base pairs; TD = tandem duplication; // = break in representation; scale bar in the bottom left corner applies to all representations). The loss of *accD* is a shared feature of the Campanulaceae, Cyphiaceae, and Lobeliaceae plastomes (Figure 1). Segments of the *rbcL*-*psaI* region are numbered for convenient reference. The inferred ancestral plastome of the Campanulaceae and Cyphiaceae retained only the even-numbered segments, and this DNA flanking their shared inversion (see Figure S1K in [4]) was likely present in the ancestral lineage of the Lobeliaceae. The common ancestral plastome also had a transposed copy of *rpl23* inserted in the intergenic region between *trnC(GCA)* and *rpoB* (Figure 5), which served as the source of the secondary transposition that incorporated the first six bp of *rpl23* into the inferred ancestral ORF200. It is not possible to determine precisely what was present between ORF200 and the *accD* remnants because all descendant lineages experienced inversions and associated deletions at this hot spot. The 3′ end of ORF200 was evidently duplicated during the inversion, and a subsequent deletion left an inverted 118-bp repeat that abuts the last vestige of *accD*. This inversion junction also has a foreign DNA segment inserted downstream of *trnK(UUU)* that is inferred to have carried ORF148 (on the opposite strand). Between *trnK(UUU)* and ORF148 is a 64-bp transposed copy from the *trnC(GCA)*-*rpoB* intergenic spacer that also matches the segment present in the secondary transposition upstream of ORF200, so it is not possible to determine the source. A 53-bp segment of intergenic DNA from the *psbA*-*trnK(UUU)* region was deleted. The inverted 118-bp copies of 3′ ORF200 permitted a stem-loop reversion in *Grammatotheca bergiana* (Figure 1) that re-established the ancestral gene order.

**Figure 4 plants-11-02863-f004:**
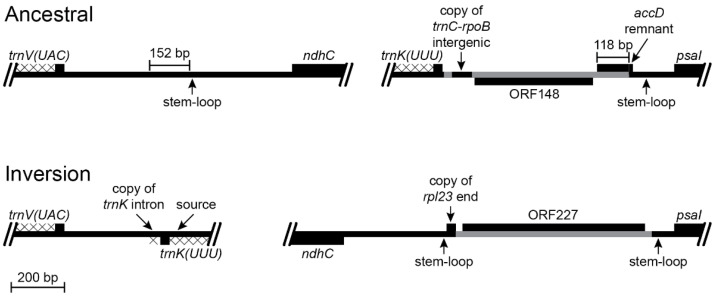
Reconstruction of the *trnV(UAC)*-*ndhC* and *trnK(UUU)*-*psaI* regions before and after an inversion and the associated insertions of foreign DNA and transposed copies of plastid DNA (plastome map depiction as in Figure 3). This inversion occurred in the common ancestral lineage of *Lobelia galpinii* and the remaining Lobeliaceae (Inversion 2 in Figure 1) and many species preserve this plastome arrangement. The small partial copy of the 3′ end of *rpl23* and the foreign DNA carrying ORF227 inserted upstream of *psaI* replaced the DNA segment previously located there. The base of the stem-loop structure in the *trnV(UAC)*-*ndhC* region was one break-point for the 152-bp deletion, and the enlarged stem after the inversion includes nucleotides of the *rpl23* fragment, which is consistent with the stem-loop ligation model (Figure 2). A small segment of the 3′ end of ORF227 has weak amino acid sequence similarity with the 118-bp remnant of ORF200, and may have played a role in recombination, but the plastid DNA was deleted up to the edge of the stem-loop structure upstream of *psaI*. The new *trnV(UAC)-trnK(UUU)* junction contains a small, duplicated segment of the *trnK(UUU)* intron, but the ligation points seem arbitrary.

**Figure 5 plants-11-02863-f005:**
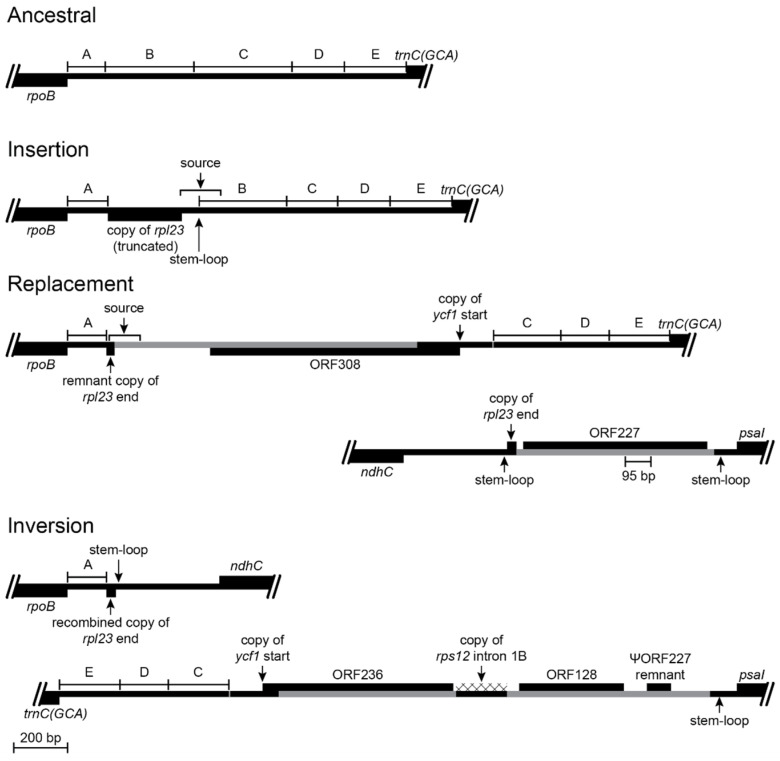
Reconstruction of the *rpoB*-*trnC(GCA)* and *ndhC*-*psaI* regions before and after an inversion and the associated insertions of foreign DNA and transposed copies of plastid DNA (plastome map depiction as in Figure 3). This inversion occurred in the common ancestral lineage of the *Lobelia fervens*-*L. laxa* clade (Figure 1), but no species are known to preserve this plastome arrangement with just these three inversions. The *rpoB*-*trnC(GCA)* intergenic region has a complex history, and the segments are lettered for convenient reference. An almost complete and potentially functional copy of *rpl23* was inserted in the common ancestral lineage of the Campanulaceae, Cyphiaceae, and Lobeliaceae (Figure 1), the insertion site shows evidence of stem-loop ligation (Figure 2), and this junction region was the source of a secondary duplicative transposition to a site downstream of *rbcL* (Figure 3). A 29-bp remnant remained after most of the *rpl23* copy and intergenic segment B were replaced by foreign DNA carrying ORF308, which incorporated a copy of the *ycf1* start. The junction of the remnant copy of *rpl23* and the inserted foreign DNA was the source for a secondary duplicative transposition to the *rps4-trnT(UGU)* region in an ancestral lineage of *L. laxa* (Figure 7). The inversion involved recombination of the *rpl23* fragments, the ancestral ORF308 was truncated and modified to what is now ORF236 in *L. laxa*, and a pseudogene remnant of ancestral ORF227 remains. A small segment of the 3′ end of ORF236 has weak amino acid sequence similarity with both ancestral ORF308 and ancestral ORF227, and may have played a role in recombination. A transposed copy of *rps12* intron 1B and foreign DNA carrying ORF128 are present in the inversion junction, but derived point mutations indicate that the *rps12* intron 1B copy was either very recently inserted in the plastome of *L. laxa* or it has undergone concerted evolution with the source region in the IR.

**Figure 6 plants-11-02863-f006:**
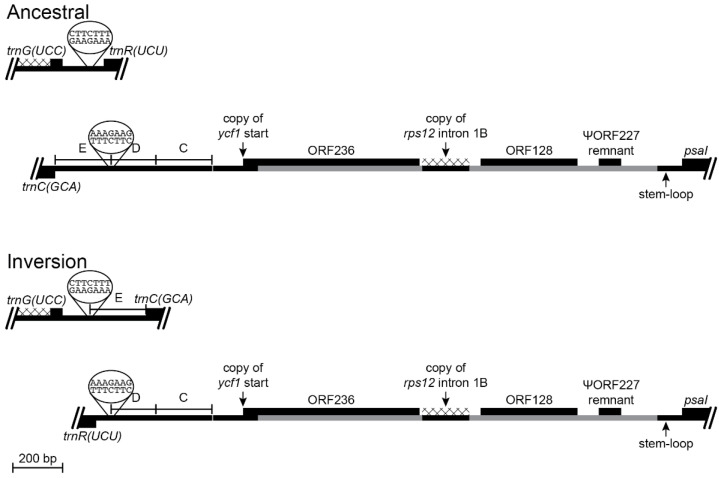
Reconstruction of the *trnG(UCC)*-*trnR(UCU)* and *trnC(GCA)*-*psaI* regions before and after an inversion (plastome map depiction as in Figure 3). This inversion occurred in the common ancestral lineage of the *Lobelia fervens*-*L. laxa* clade (Figure 1) and is preserved in *L. laxa*. Recombination occurred at inverted seven-bp segments that were present in the ancestral plastome, with no associated insertions or deletions.

**Figure 7 plants-11-02863-f007:**
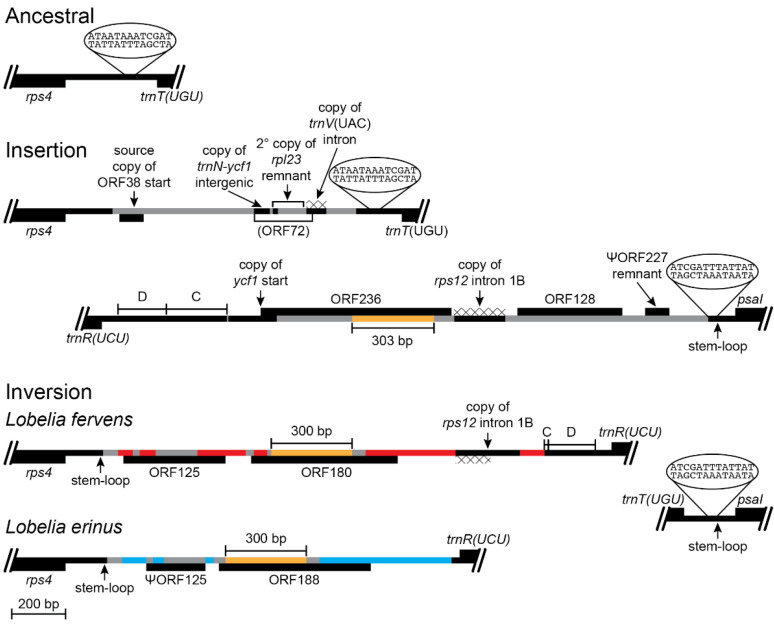
Reconstruction of the *rps4-trnT(UGU)* and *trnR(UCU)-psaI* regions before and after an inversion and the associated insertions of foreign DNA and transposed copies of plastid DNA (plastome map depiction as in Figure 3). This inversion occurred in the common ancestral lineage of the *Lobelia fervens*-*L. erinus* clade (Figure 1) and is preserved in both species. The *rps4-trnT(UGU)* region in *L. laxa* has the DNA insertions shown, and no intergenic plastid DNA was deleted at the insertion site. Only *L. laxa* has ORF38 inserted in the highly conserved *trnL(CAA)-ndhB* region of the IR, but endpoint analysis indicates the copy in the *rps4-trnT(UGU)* region was the source of the duplicative transposition, with the end of this original copy subsequently deleted. As a result, it is not possible to determine exactly when ORF38 was originally inserted in the *rps4-trnT(UGU)* region. It is also not possible to determine the age of the small copy of the *trnN(GUU)-ycf1* intergenic region, but the duplicated copy of the *rpl23* remnant with the flanking foreign DNA from the *rpoB-trnC(GCA)* region (Figure 5) indicates that this secondary transposition occurred before the source region was deleted from the common ancestral plastome of the *Lobelia fervens*-*L. laxa* clade. The transposed copy of the *trnV(UAC)* intron is also relatively ancient, as evidence by the absence of four derived point mutations now present in the contemporary source region in *L. laxa*. The heterogeneous composition of this segment makes the potential ORF72 almost certainly spurious. Whatever inserted DNA was present in the *rps4-trnT(UGU)* region was evidently deleted during the inversion, along with 68 bp of the flanking intergenic plastid DNA. One inversion junction formed by recombination of a 13-bp segment located downstream of *trnT(UGU)* that coincidentally matched one end of the large stem-loop structure located upstream of *psaI*. The other junction has a newly formed stem-loop structure terminating the plastid intergenic DNA upstream of *rps4* and DNA insertions that replaced much of the foreign DNA and transposed copies of plastid DNA that previously flanked block C from the *rpoB*-*trnC(GCA)* region. A 300-bp core (colored gold) in ORF180 (*L. fervens*) and ORF188 (*L. erinus*) corresponds to a central segment from ORF236 (*L. laxa*), but the flanking regions have experienced serial replacement. Despite the similarity in size and position, ORF128 in *L. laxa* has no significant similarity to ORF125 in *L. fervens*. The copies of *rps12* intron 1B are transposed from overlapping source regions, but they are inserted in different locations and both have derived mutations indicative of independent transpositions or concerted evolution with the corresponding source regions. The DNA trafficking at this site is also apparent in the comparison of *L. fervens* and *L. erinus*. In addition to the shared 300-bp core, the foreign DNA shared exclusively by these two species is marked in gray, while the unique segments are marked in red and blue, respectively.

## Data Availability

All sequenced plastomes are available at NCBI, with GenBank accession numbers AJ316582, AY582139, DQ383815-16, DQ898156, EU090187, KT372780, KY354213-29, MF770602-35, Z00044.

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
