# Peer review of "DNA Sequence Analysis of an Inversion Hot Spot in Lobeliaceae Plastomes"

_plants, 2022, doi:10.3390/plants11212863_

Round 1
Reviewer 1 Report
The paper "DNA Sequence Analysis of an Inversion Hot Spot in Lobeliaceae Plastomes "reconstructs the underlying molecular factors responsible for five inversions in Lobeliaceae plastomes. Four of the five inversions documented in the paper are consistent with the disruption rescue model, with one endpoint located at the inversion hot spot located at the accD deletion site. The "accD deletion site" describes what is not present, not what is present. This inversion hot spot is characterized by serial replacement of foreign DNA at a site upstream of psaI adjacent to a large stem-loop structure. This article further develops and enriches some of the information presented in the author's previous article: The dynamic history of plastid genomes in the Campanulaceae sensu lato is unique among angiosperms.
Remarks:
Figure 3 and figure 4 are mentioned in the text way ahead of the figures in the article!
Line 97 – "At the smallest possible scale, hairpin inversions are common [12]; see also [4] (Figure S3B), which occur because the flanking sequences are palindromic (comprising reverse and complementary nucleotides) and can form miniature stem-loop structures." - see also… it is not clearly written where.
Reviewer 2 Report
This paper performed a comprehensive comparative genome analysis to investigate the structure and organization in Lobeliaceae plastome, which provided the useful information to better understand the evolution of plant chloroplast genome as well as contribute to phylogenetic relationship in Lobelia. In my opinion, this paper is well prepared and is suitable to publish in plants. I have only some small concerns.
1. Whether is it possible to validate the inversion or duplication events by PCR?
2. It is better to reduce the keywords. I think 9 keywords is too cumbersome
3. The methods section should be more informative and detailed.
